# Loss of miR-1469 expression mediates melanoma cell migration and invasion

**Mallory J. DiVincenzo[1,2], Zoe Barricklow[2], Emily Schwarz[3], Maribelle Moufawad[2], J. Harrison Howard[2], Lianbo Yu[2], Catherine Chung[2], Alejandro A. Gru[4], William E. Carson, III[2]***

**1** Department of Veterinary Biosciences, The Ohio State University, Columbus, Ohio, United States of America, **2** The Arthur G. James Cancer Hospital and Solove Research Institute, The Ohio State University, Columbus, Ohio, United States of America, **3** Biomedical Sciences Graduate Program, College of Medicine, The Ohio State University, Columbus, Ohio, United States of America, **4** Department of Pathology, University of Virginia, Charlottesville, Virginia, United States of America

* william.carson@osumc.edu

**Data Availability Statement:** All relevant data are within the manuscript and its Supporting information files.

## Abstract

Tumor ulceration is considered one of the most prognostically significant findings in primary cutaneous melanoma, associated with decreased disease-free and overall survival. However, the unique features associated with ulcerated melanoma that contribute to a poor prognosis in affected patients remain poorly defined. microRNAs are small, non-coding RNAs that function to inhibit expression of specific gene targets, therefore altering the functions of cells in which they are expressed. miR-1469 is a novel miR with significantly decreased expression in ulcerated melanoma tissue relative to non-ulcerated tumors. We hypothesized that loss of miR-1469 expression in melanoma contributes to altered tumor cell functions mediating disease progression. Transfection of a miR-1469 mimic resulted in a significant reduction in the migratory and invasive capacity of the CHL1 and MEL39 melanoma cell lines (>58.1% reduction, p < 0.0332), as well as the invasive capacity of the A375 melanoma cell line (>50% reduction, p < 0.0021). Expression of myeloid cell leukemia-1 (MCL1), a miR-1469 target gene, was reduced in the A375 and MEL39 cell lines by immunoblot. No significant differences in viability, resistance to apoptotic stimuli, or proliferation were observed following transfection. These findings together demonstrate how migration and invasion are specific functions through which miR-1469 expression in melanoma cells can contribute to the differences in disease progression associated with tumor ulceration.

## Introduction

Melanoma is the leading cause of skin cancer-related death, associated with nearly 7,000 deaths in 2020 in the United States alone [1]. Cutaneous melanoma can sometimes present with ulceration of the tumor surface. Tumor ulceration is defined as the full thickness loss of epidermis overlying the tumor to the level of the basement membrane, with evidence of a host inflammatory response and associated thinning, effacement, or reactive hyperplasia of the adjacent epidermis [2]. Tumor ulceration has been correlated with both decreased disease-free

**Funding:** National Institutes of Health grant T32 CA009338 awarded to MJD. The funders had no role in study design, data collection, analysis, decision to publish, or preparation of the manuscript.

**Competing interests:** The authors have declared no competing interests exist.

and overall survival in melanoma patients with otherwise similar staging criteria [3, 4]. Therefore, recognition of gross or histologic evidence of ulceration in cutaneous melanoma tissue has been incorporated into melanoma staging to better stratify melanoma patients [5, 6]. While ulceration is a feature that provides prognostic information to clinicians, few studies have explored the molecular features that explain its poor prognosis. Therefore, further studies are needed to characterize the molecular pathways that drive the ulcerated phenotype.

microRNAs (miRs) are small, non-coding RNA molecules 21–23 nucleotides long that function to regulate the expression of specific homologous gene targets via degradation of target gene mRNA or inhibition of protein translation [7]. Multiple miRs demonstrate consistent patterns of dysregulated expression in the setting of cancer [8–11]. Dysregulated expression of specific miRs leads to altered tumor cell function via miR-mediated regulation of gene expression [7, 10]. Our group has a longstanding interest in characterization of miR expression within unique presentations of melanoma, including cutaneous melanoma presenting with tumor ulceration. Reduced expression of miR-1469 was identified in our NanoString assessment of miR expression patterns in ulcerated melanomas when compared to non-ulcerated tumors. miR-1469 has also demonstrated consistently downregulated expression in other tumor types such as in lung, laryngeal and esophageal squamous cell cancers [12–14]. Furthermore, low expression of miR-1469 in esophageal squamous cell cancer was found to correlate with lymph node metastasis, tumor invasiveness and worsening disease progression [14]. Overexpression of miR-1469, on the other hand, was found to inhibit expression of myeloid cell leukemia-1 (MCL1) and consequently promote apoptosis in laryngeal cancer cells. Additionally, increased expression of the tumor suppressor gene p53 was found to promote miR-1469 production [13]. Thus, given that miR-1469 is not only involved in regulation of these few cancer-associated genes, but has a total of 75 possible target genes, it is likely that miR-1469 may be influential in tumorigenesis across many other cancer types as well [15]. While abnormal miR-1469 expression has indeed been identified in multiple malignancies, neither a role for miR-1469 nor the effects of its reduced expression in melanoma have been previously reported. In the present study, the functional effects of miR-1469 on melanoma biology were examined by evaluating proliferation, migration, invasion and survival *in vitro* when miR levels were experimentally manipulated. These studies showed that diminished levels of miR-1469 promoted increased melanoma cell migration and invasion and implicated regulation of miR-1469 target gene myeloid cell leukemia-1 (MCL1) as a possible mechanism for this effect [13, 16–20].

## Materials and methods

### Patient samples and RNA isolation from formalin fixed paraffin embedded tissue

Four 20 μm thick scrolls of formalin fixed, paraffin embedded tissue from biopsy samples of ulcerated primary cutaneous melanoma and non-ulcerated primary cutaneous melanoma were collected for use under an approved IRB protocol (2007C0015). Under this IRB protocol, a waiver of patient consent was approved as the data was analyzed anonymously. Samples for inclusion were selected and classified as ulcerated (n = 13) or non-ulcerated (n = 11) by two dermatopathologists who were blinded as to the prior classification. RNA was isolated using the Invitrogen RecoverAll Nucleic Acid Isolation kit, quantified, and stored and assessed for microRNA expression using the NanoString human miRNA assay, as previously described, and by qPCR, as described below [21, 22].

## Cell lines

The A375, MEL39, and CHL1 melanoma cell lines were purchased from the ATCC. A375 and CHL1 were cultured in Dulbecco's Modified Eagles Medium (DMEM) supplemented with 10% fetal bovine serum (FBS). MEL39 was cultured in RPMI supplemented with 10% FBS. All cell lines were cultured and maintained in 5% $CO_2$ at 37˚C.

## Transfections

Cells were transfected with a mirVana microRNA mimic (#MC14120, Invitrogen, Carlsbad, CA) of hsa-miR-1469 (mature miRNA sequence: CUCGGCGCGGGGCGCGGGCUCC) or a corresponding negative control miR-mimic scramble construct (#4454059, Invitrogen) using Mirus TransIT TKO Transfection Reagent (Mirus Bio, Madison, WI) in Opti-Modified Eagles Medium according to the manufacturer's protocol at a final concentration of 25 nM for a minimum of 24 hours prior to use in all assays.

## RNA isolation and quantitative real-time PCR

Total RNA was isolated from melanoma cell lines 24 hours following transfection using Trizol reagent (Invitrogen) according to manufacturer's instructions and quantified using a Nano-Drop spectrophotometer (ThermoFisher). RNA concentration was normalized across samples prior to universal synthesis of cDNA using the TaqMan Advanced miRNA cDNA Synthesis kit according to manufacturer's instructions (Invitrogen). qPCR was then performed using the TaqMan Fast Advanced Master Mix (ThermoFisher, #4444556) and 7900HT Fast Real-Time PCR System (Applied Biosystems) with the TaqMan Advanced miRNA assay for hsa-miR-1469 (ThermoFisher, TaqMan Assay ID # 480771). The TaqMan Advanced miRNA assay for hsa-miR-16 (ThermoFisher TaqMan Assay ID # 477860) was used as an endogenous control. Relative microRNA expression was determined using the ΔΔCt method [23].

## Migration and invasion assays

Uncoated and Matrigel-coated Transwell culture systems (0.8 um pore size) were employed to assess the migratory and invasive capacity of tumor cells, respectively (Corning, Corning, NY). Following transfection, tumor cells were collected and plated on the upper face of a Transwell insert at a concentration of 1 x $10^5$ cells/mL in DMEM or RPMI, each supplemented with 0.2% FBS. Inserts were then placed into wells containing DMEM or RPMI supplemented with 10% FBS as a chemoattractant. After culture of cells for 24 hours at 37˚C and 5% CO2, cells remaining on the upper face of the transwell inserts were mechanically removed and adherent cells on the lower face of the inserts were fixed and stained using Dip Quick cytological stains (Jorgensen Laboratories, Loveland, CO). Each insert was then evaluated by phase-contrast microscopy and digitally photographed using an EVOS Cell Imager (ThermoFisher). Five representative images at 20x magnification were analyzed for each insert. ImageJ software was used to quantify relative transwell migration or invasion according to a published protocol [24].

## Proliferation assays

Proliferation of melanoma cell lines following 24 hours of transfection with a miR-1469 mimic or miR scramble construct was measured as absorbance at 490 nm using the CellTiter 96 Aqueous (MTS) Cell Proliferation Assay kit at 24 and 48 hours following transfection according to manufacturer's instructions (Promega).

## Viability and apoptosis assays

Viability of cells following transfection was determined using trypan blue staining and quantified using an EVE Automated cell counter (Nanoentek, Waltham, MA) and calculated as the percentage of live cells per mL relative to total cells per mL. Cells were also assessed for altered susceptibility to apoptosis following over-expression of miR-1469. 24 hours following transfection, cells were treated with 1uM staurosporine for increasing periods of time (0, 1, 3, 16, and 24 hours). Cells were then collected and stained with Annexin V-APC (BD Pharmingen #550474) and Propidium iodide-PE (BD Pharmingen #556463) to assess for apoptosis-induced phosphatidylserine exposure and cell membrane integrity. Heat-treated cells were used as a positive control. Unstained cells as well as Annexin V-only and propidium iodide-only stained cells were used as single-color controls. Annexin V and Propidium iodide staining was assessed using an LSR II Flow cytometer (BD Biosciences). Annexin V positivity was analyzed using FlowJo software [25].

## Immunoblot analysis

48 hours following miR transfection, cells were washed with PBS and lysed with Pierce RIPA buffer containing Halt protease and phosphatase inhibitors (ThermoFisher). Lysates were incubated on ice for 30 minutes prior to centrifugation and removal of cell debris and stored at -20˚C. Protein concentration was determined using the Bradford Reagent Assay Kit (BioRad) according to manufacturer's instructions. Protein samples were then denatured in Lammeli buffer (BioRad) and boiled for 5 minutes prior to separation by SDS-PAGE. Proteins were then transferred onto a nitrocellulose membrane (BioRad). After blocking with a 5% BSA solution, the membrane was incubated overnight at 4˚C with primary antibody, then washed prior to incubation with fluorescent secondary antibodies (LiCor, 1:5000 dilution). Primary antibodies were purchased from Proteintech (MCL1, #16225-1-AP, 1:1000 dilution, and GAPDH, #60004-1-Ig, 1:5000 dilution). Membranes were imaged using the LiCor Odyssey CLX system. Detected signal was quantified and normalized to background and loading controls using the LiCor ImageStudio software according to manufacturer's instructions.

## Statistical analysis

All statistics for *in vitro* data were performed using GraphPad Prism 9 statistical software. For all *in vitro* assays, statistical significance of differences between groups was analyzed using ANOVA or two-tailed Student's *t* test and all data is based on 3 experimental replicates. Additionally, statistical software SAD 9.4 and R 3.6 was used for Nanostring data analysis with a fold change of at least 1.5 for any differentially expressed miRs identified. A p-value less than or equal to 0.05 was considered to be statistically significant. The Holm-Bonferroni method was used to adjust for multiple comparisons.

# Results

## miR-1469 expression is significantly decreased in ulcerated cutaneous melanoma relative to non-ulcerated cutaneous melanoma

miR-1469 expression was found to be decreased in a group of ulcerated primary cutaneous melanoma samples relative to non-ulcerated cutaneous melanoma by Nanostring miR analysis (1.34 fold decrease, p = 0.0482, Fig 1A). In order to confirm the differential pattern of miR-1469 expression observed by Nanostring between ulcerated and non-ulcerated melanoma tumors, qPCR was performed to assess miR-1469 expression. miR-1469 was significantly decreased among cutaneous melanoma samples exhibiting ulceration relative to non-ulcerated

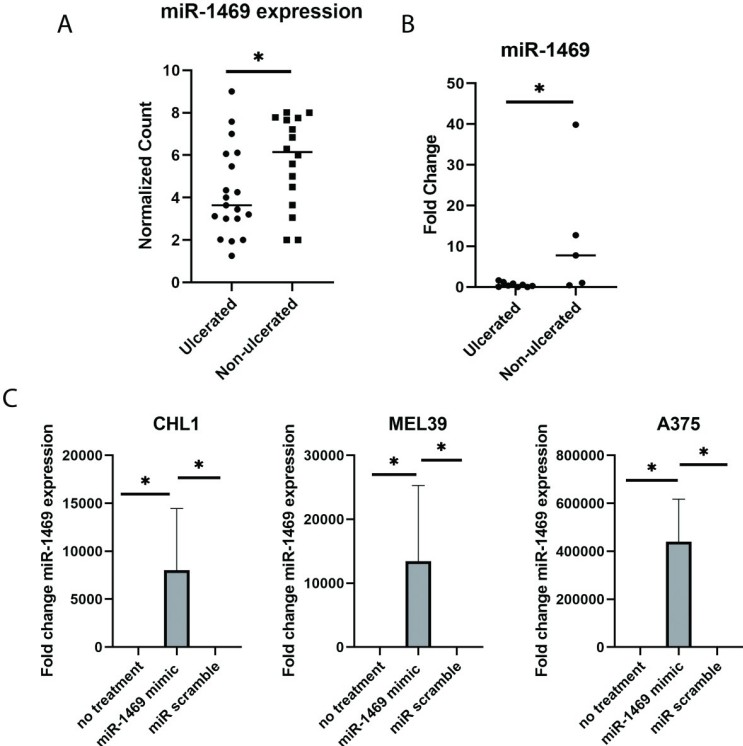

**Fig 1. miR-1469 expression in melanoma tissue and melanoma cell lines.** (A) Normalized expression level of miR-1469 in FFPE tissue in ulcerated tumors relative to non-ulcerated tumors, determined by Nanostring (1.34 fold change, p = 0.0482). (B). miR-1469 expression in FFPE tissue from ulcerated primary cutaneous melanomas relative to non-ulcerated tumors, as assessed by qPCR (11.81 mean fold change, p = 0.043). (C) qPCR for miR-1469 expression following transfection in total RNA isolated from CHL1, MEL39, and A375 cell lines after 24 hours of transfection (* = p <0.05).

cutaneous melanoma tissue, as determined by qPCR (11.81 mean fold change in expression, p = 0.043, Fig 1B).

## Expression of miR-1469 is significantly increased in melanoma cell lines upon transfection with a miR mimic

As miR-1469 dysregulation was found to be a feature associated with ulcerated primary cutaneous melanoma and limited studies have explored its role in cancer, an *in vitro* functional assessment of miR-1469 expression was performed using melanoma cell lines. Multiple melanoma cell lines were assessed for basal expression of miR-1469 by qPCR. miR-1469 expression was not detected in the CHL1, MEL39, or A375 melanoma cell lines. This result parallels the previous finding of relatively decreased expression of miR-1469 in ulcerated cutaneous melanoma tumors and supported use of these cell lines in *in vitro* assays to investigate how restoration of miR-1469 expression might affect cellular functions. This assessment was performed following transfection of melanoma cell lines with a miR-1469 mimic construct with comparisons being made to the effects of a nonspecific, commercially available negative control miR scramble construct and untransfected cells. At 24 hours following transfection, the expression of miR-1469 was significantly increased in all three cell lines relative to both untransfected and miR scramble transfected controls (p<0.0467, Fig 1C).

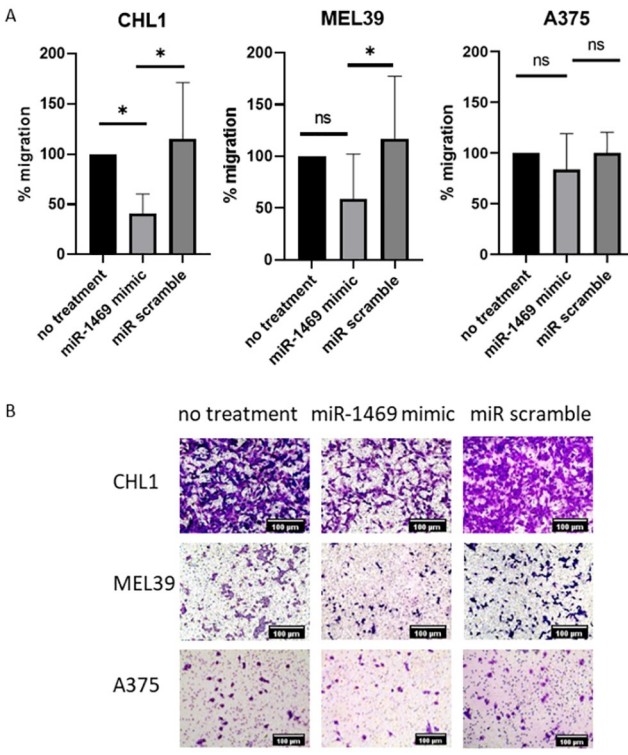

**Fig 2. miR-1469 expression reduces migration of melanoma cells.** (A) Transwell migration by melanoma cell lines CHL1, MEL39, and A375 when transfected with miR-1469, expressed as percent migration relative to untreated cells (no treatment). * = p<0.0332. (B) Representative images of Transwell migration assay, imaged at 20x magnification.

## Expression of miR-1469 alters the migratory capacity of melanoma cell lines *in vitro*

Ulceration is used as a criterion for disease staging in melanoma in part due to the association between ulceration, reduced disease-free survival, and increased invasion of tumor cells [26]. Therefore, cellular functions that contribute to disease spread and metastasis, including migration and invasion, were assessed to determine if expression of miR-1469 played a role in mediating these functions. The impact of miR-1469 expression on the ability of melanoma cell lines to migrate toward a stimulus (a serum gradient) was assessed using a transwell migration assay. CHL1 cells transfected with a miR-1469 mimic demonstrated a 58.8% decrease in migration relative to untransfected cells (p = 0.0463), and a 74.0% decrease in migration relative to CHL1 cells transfected with a miR scramble construct (p = 0.0132, Fig 2). A similar pattern of decreased migration with miR-1469 expression was observed in the MEL39 cell line, with a 58.1% reduction in migration being achieved when cells transfected with a miR-1469 mimic were compared to those transfected with control (miR scramble, p = 0.0214). While this pattern was also observed for the migration of A375 cells upon transfection, significant differences were not observed between treatments.

## Expression of miR-1469 alters the invasive capacity of melanoma cell lines *in vitro*

Next, the effects of miR-1469 expression on the invasive capacity of melanoma cell lines was evaluated using a Matrigel-coated transwell invasion assay. After incubation for 24 hours, the

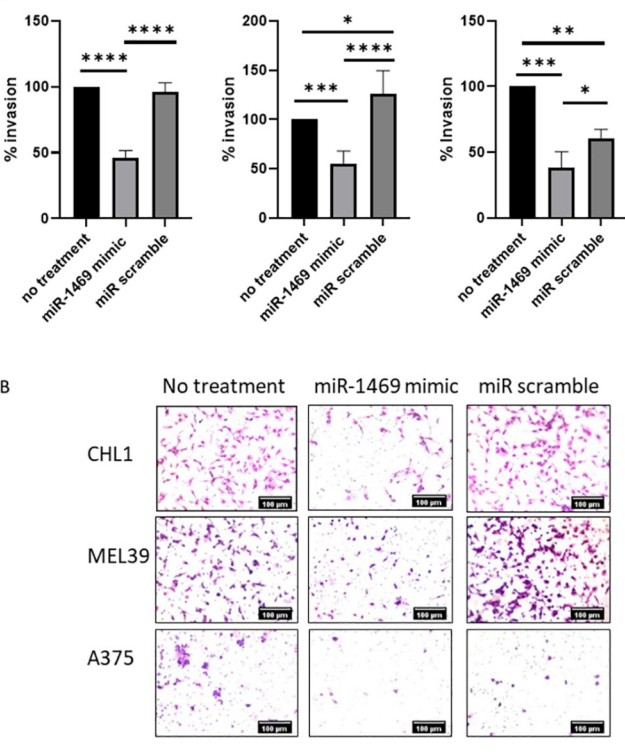

**Fig 3. miR-1469 expression inhibits melanoma cell invasiveness.** (A) Relative percent invasion of CHL1, MEL39, and A375 melanoma cell lines when normalized to untreated cells using Matrigel transwell invasion assays. (* = p<0.0332, ** = p<0.0021, *** = p<0.0002, **** = p<0.0001) (B) Representative images at 20x magnification of Matrigel invasion assay for each cell line and treatment condition.

CHL1 cell line transfected with miR-1469 exhibited a significant reduction in invasion through Matrigel matrix relative to both untransfected cells (54% reduction, p <0.0001) and miR scramble transfected cells (50% reduction, p <0.0001, Fig 3). This effect of miR-1469 expression on the invasive capacity of tumor cells was also observed in the other two cell lines. For the MEL39 cell line, miR-1469 transfection resulted in a 44.8% reduction in invasion relative to untransfected cells (p = 0.0004) and a 71.7% reduction in invasion relative to miR scramble transfected cells (p < 0.0001). In A375 cells there was a 61.6% reduction in invasion relative to untransfected cells (p = 0.0002) and a 22.3% reduction in invasion relative to the miR scramble transfection (p = 0.0322).

## Expression of miR-1469 does not significantly impact the viability of melanoma cell lines

The previous assays demonstrated a clear effect of miR-1469 on cellular migration and invasion. In order to confirm whether these effects may be due to increased cell death upon transfection with the miR-1469 mimic construct, the viability of each cell line was assessed by trypan blue staining at 48 hours following transfection, equivalent to the time point at which migration and invasion were assessed. For all three cell lines, viability remained above 90% regardless of treatment, and there were no significant differences in cell viability between groups (Fig 4A).

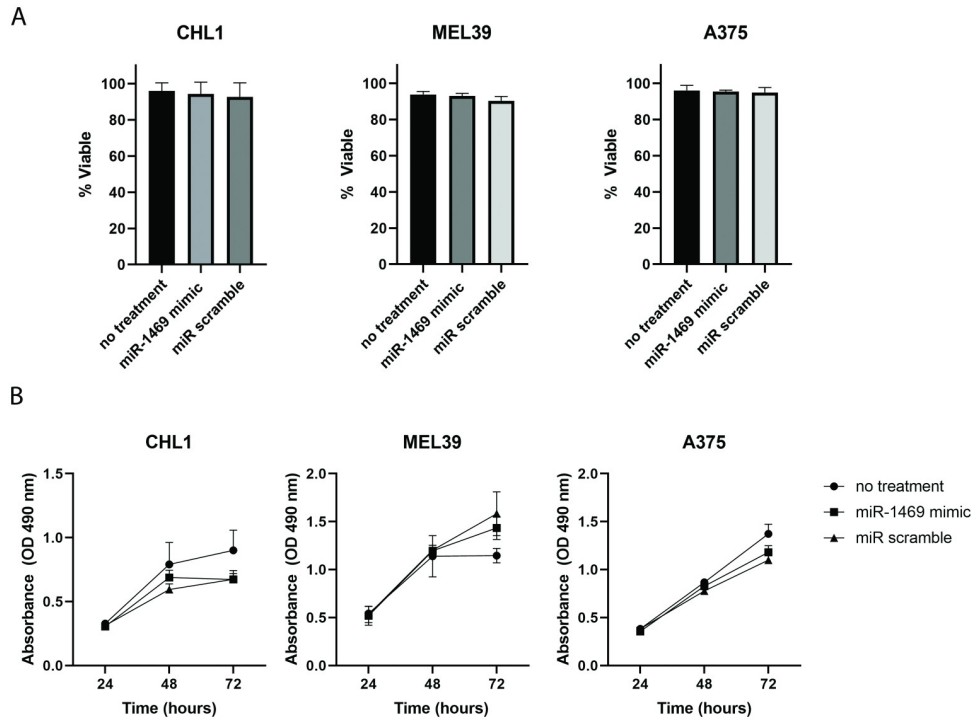

**Fig 4. miR-1469 transfection of melanoma cell lines does not alter viability or proliferation.** (A) Trypan blue staining for assessment of viability following transfection of melanoma cell lines with miR-1469 relative to scramble-transfected and untransfected cells at 48 hours. (B) MTS Proliferation assays performed at 24, 48, and 72 hours following transfection of melanoma cell lines with miR-1469 miR relative to scramble-transfected or untransfected cells.

## Expression of miR-1469 does not impact proliferation of melanoma cell lines

Tumor ulceration in melanoma has previously been shown to correlate with mitotic rate, another prognostic criterion used as a means to assess proliferative rate in melanoma staging. Thus, increased tumor cell proliferation may be present in the setting of ulceration, and it was hypothesized that miR-1469 over-expression might affect the proliferation of melanoma cell lines [27, 28]. However, there was no significant difference in the proliferation of miR-1469-transfected CHL1, MEL39, or A375 cells at 24, 48, and 72 hours relative to untransfected or miR scramble transfected controls, as determined by MTS proliferation assay ($p = 0.41$, 0.68, and 0.78 for CHL1, MEL39, and A375, respectively, Fig 4B).

## Susceptibility to apoptosis is not significantly altered with miR-1469 expression

In order to determine whether miR-1469 expression plays a role in protecting melanoma cells from apoptotic cell death, melanoma cell lines with increased expression of miR-1469 were exposed to staurosporine, a protein kinase inhibitor known to induce apoptosis via both caspase-dependent and independent pathways [29]. There was no significant difference in the fraction of apoptotic cells in the CHL1, MEL39, or A375 cell line following mir-1469 transfection and staurosporine treatment as compared to controls (minimum p-value = 0.153, 0.102, and 0.119 for CHL1, MEL39, and A375, Fig 5A and 5B). While the CHL1 cell line did exhibit a

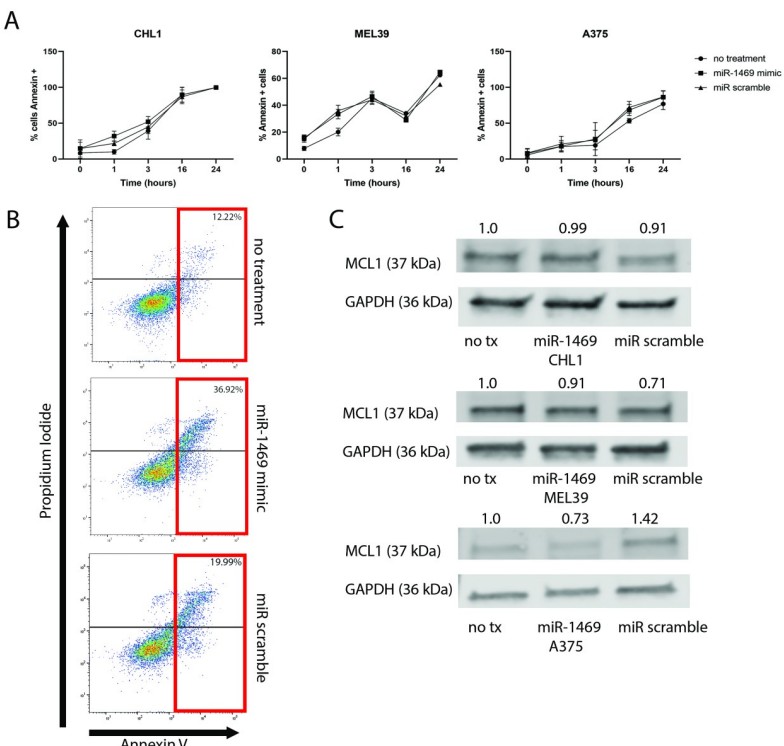

**Fig 5. miR-1469 expression in melanoma does not significantly alter susceptibility of tumor cells to apoptosis via regulation of MCL1.** (A) 24 hours after melanoma cell lines had been transfected with miR-1469 or a scramble miR or left untransfected, they were treated with 1 uM staurosporine for 0, 1, 3, 16, or 24 hours and assessed for apoptosis as determined by flow cytometry for Annexin V positive cells. (B) Representative flow cytometric scatter plots depict percent Annexin V positive cells (x axis) in CHL1 cells following 1 hour exposure to 1uM staurosporine. The percentage of Annexin V positive cells is indicated in red. (C) Protein expression of MCL1 in CHL1, MEL39, and A375 cell lines 72 hours following transfection, determined by immunoblot. GAPDH was used as a loading control. Densitometry indicated above as a ratio relative to untreated (no tx) cells for each cell line, normalized to GAPDH.

24.7% increase in Annexin V+ cells relative to untransfected cells after 1 hour of treatment, this trend was not sustained at later time points and was not found to be statistically significant.

## miR-1469 significantly alters the expression of MCL1 in melanoma *in vitro*

In addition to its well-recognized role as an inhibitor of apoptosis, MCL1 has also been shown to have an effect on the migratory and invasive capacity of tumor cells [17]. Given its increased expression in melanoma and its confirmation as a gene target for miR-1469 in laryngeal cancer, we assessed whether MCL1 may also function as a target gene in the context of melanoma [13, 20]. In order to determine this, the expression of MCL1 following miR-1469 transfection of melanoma cell lines was assessed by immunoblot to determine whether it may function as a target gene in the context of melanoma. At 72 hours following miR transfection, expression of MCL1 was decreased in the A375 and MEL39 melanoma cell lines (Fig 5C).

## Discussion

Ulcerated primary cutaneous melanoma lacks a well-established biological basis for the differences observed in clinical behavior in affected patients. In the current study we have

demonstrated that expression of miR-1469 is consistently downregulated across a set of ulcerated primary cutaneous melanoma tumors relative to non-ulcerated tumors. miR-1469 has not been previously investigated for its role in regulating the functions of melanoma cells. The current study demonstrated that miR-1469 over-expression specifically inhibited the migration and invasive capacity of melanoma cell lines. Therefore, downregulation of miR-1469 could contribute to disease progression and tumor metastasis in melanoma.

Use of a transfection model to achieve expression of miR-1469 did not demonstrate a significant effect on the viability of melanoma cells. This finding supports a model in which the observed effects on migration and invasion were due to the interaction of mir-1469 with functional target genes. The lack of an impact of miR-1469 on tumor cell apoptosis following exposure to staurosporine supports the notion that increased levels of miR-1469 did not serve as a noxious stimulus in the context of the current experiments.

Both the migration and invasion of multiple melanoma cell lines was significantly inhibited when miR-1469 was overexpressed *in vitro*. These findings indicate a role for miR-1469 in negatively regulating these processes in melanoma via interaction with target genes. Loss of miR-1469 expression may therefore act to promote melanoma cell migration and invasion of adjacent tissues in the setting of ulceration. Promotion of such features that contribute to malignant behavior when miR-1469 is low in tumor tissue aligns with findings of a previous study in which low expression of miR-1469 was correlated with depth of invasion, presence of lymph node metastasis, pathological tumor stage, and poorer disease-free and overall survival in patients with esophageal squamous cell cancer [14]. However, the present results also stand in contrast to a recent study of miR-1469 as a promoter of cellular invasion in pancreatic cancer cell lines via inhibition of metastasis suppressor gene NDRG1 with subsequent activation of NF-κB [30]. While NF-kB activation has also been shown to promote metastasis and invasiveness in melanoma through downstream activation of several potentially tumorigenic factors including osteopontin (OPN) and matrix mellanoproteinases (MMPs) this opposing result of miR-1469 function in pancreatic cancer is most likely due to the diverse and cancer context-dependent effects of microRNA biology and thus these targets were not assessed in the current study [31, 32].

As a validated target gene for miR-1469, MCL1, a pro-apoptotic protein, is susceptible to inhibition of gene expression via binding of miR-1469 within the 3'UTR of the MCL1 mRNA transcript [13]. However, transfection with miR-1469 resulted in an inhibitory effect on MCL1 protein expression that was variable, depending on the melanoma cell line examined. Differences in the mutational profiles of melanoma cell lines may impact the observed effects of miR-1469 expression and its impact on MCL1 as a target gene. For example, the CHL1 line is $BRAF^{wt}$, and $NRAS^{wt}$ and may exhibit a different degree of dependence on MCL1 expression relative to MEL39 and A375, which both carry the $BRAF^{V600E}$ mutation. Recent studies have demonstrated dependence on MCL1 expression for the resistance of BRAF$^{V600E}$ mutant melanomas to apoptotic stimuli via the constitutive activation of the RAS/RAF/MEK/ERK signaling pathway. These signaling events lead to the phosphorylation and stabilization of MCL1 [20, 33]. Also, the MEL39 cell line harbors a truncating mutation in PTEN, which may result in its inactivation and loss of PTEN-mediated expression of proapoptotic machinery including caspases and BID [34]. The effect of miR-1469 on MCL1 and sensitivity to apoptosis can thus be modulated by factors emanating from common melanoma mutations.

MCL1 has also recently been shown to mediate tumor cell migration via binding anion channels on the mitochondrial membrane, resulting in increased production of reactive oxygen species that drive migration in lung cancer [17]. MCL1 has also been shown to promote tumor invasiveness via interaction with Cofilin 1, a cytoskeletal remodeling protein, and phosphorylated SRC [19]. Therefore, the reduction in the migratory and invasive capacity observed

in melanoma cell lines when miR-1469 is overexpressed could potentially be attributed to the effects of MCL1 on these pathways.

There are a few limitations to the current study. There is a substantial difference between the miR-1469-fold change values from Nanostring versus those obtained by qPCR, however this is not unexpected. One of the main differences between Nanostring and qPCR is the fact that Nanostring lacks an amplification step. Therefore, Nanostring analysis provides direct quantification of the number of each miR of interest in a sample and minimizes error that can come with additional sample handling in the conduct of molecular assays. The amplification step can incite bias within the sample in the form of the cDNA transcripts that are used for target detection by PCR. Additionally, although MCL1 is a defined target gene of miR-1469 that holds clinical significance in the context of melanoma, the results of this study demonstrate that MCL1 expression is modestly impacted by miR-1469 alone in the melanoma cell lines examined. Therefore, definition of additional target genes for miR-1469 and examination of their functional roles in the context of melanoma is needed. Also, while clear effects on cell migration and invasion are apparent, additional investigation using co-culture systems and *in vivo* models is needed to determine how miR-1469 expression in melanoma cells impacts interaction of tumor cells with other components of its microenvironment, as well as the development of metastases.

The current study demonstrates how miR-1469 can serve as a tumor-suppressing miR in the context of melanoma and supports the loss of miR-1469 expression as a feature that promotes the biologic behavior of ulcerated melanomas. Given the accessible nature of primary cutaneous melanomas, intratumoral therapy with a miR-1469 mimic may hold promise in the treatment of ulcerated tumors. Future investigations will focus on the validation of additional miR-1469 target genes in the context of ulcerated melanoma.

## Supporting information

**S1 Fig. Representative images of A375 trypan blue staining from Fig 4.**
(TIF)

**S2 Fig. Raw images of included immunoblots from Fig 5.**
(TIF)

## Author Contributions

**Conceptualization:** Mallory J. DiVincenzo, J. Harrison Howard, William E. Carson, III.

**Data curation:** Mallory J. DiVincenzo, Zoe Barricklow, Emily Schwarz, Maribelle Moufawad.

**Formal analysis:** Mallory J. DiVincenzo, Zoe Barricklow, Maribelle Moufawad, Lianbo Yu.

**Investigation:** Mallory J. DiVincenzo, Zoe Barricklow, William E. Carson, III.

**Methodology:** Mallory J. DiVincenzo, Zoe Barricklow, Lianbo Yu, Alejandro A. Gru, William E. Carson, III.

**Project administration:** Mallory J. DiVincenzo, J. Harrison Howard, William E. Carson, III.

**Resources:** Catherine Chung, Alejandro A. Gru, William E. Carson, III.

**Supervision:** Mallory J. DiVincenzo, Lianbo Yu, William E. Carson, III.

**Validation:** Mallory J. DiVincenzo, Zoe Barricklow.

**Visualization:** Mallory J. DiVincenzo, William E. Carson, III.

**Writing – original draft:** Mallory J. DiVincenzo, Zoe Barricklow.

**Writing – review & editing:** Mallory J. DiVincenzo, Emily Schwarz, Maribelle Moufawad, J. Harrison Howard, Catherine Chung, Alejandro A. Gru, William E. Carson, III.

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
