## [Decision Letter · Decision Letter 0]

10 Jun 2021

PONE-D-21-15478

Loss of miR-1469 expression mediates melanoma cell migration and invasion

PLOS ONE

Dear Dr. Carson,

Thank you for submitting your manuscript to PLOS ONE. After careful consideration, we feel that it has merit but does not fully meet PLOS ONE’s publication criteria as it currently stands. Therefore, we invite you to submit a revised version of the manuscript that addresses the points raised during the review process.

Please respond to all critique, point-by-point. In particular:

Clarify the introduction section as suggested by both referees

Explain why you have employed hsa-miR-16 as control

Importantly: Explain the divergent fold-change values obtained by the two methods Nanostring and qPCR

provide more details on the statistics, as suggested by referee 2

We look forward to receiving your revised manuscript.

Kind regards,

Klaus Roemer

Academic Editor

PLOS ONE

Journal Requirements:

5. Please include captions for your Supporting Information files at the end of your manuscript, and update any in-text citations to match accordingly. Please see our Supporting Information guidelines for more information: http://journals.plos.org/plosone/s/supporting-information

Reviewers' comments:

Reviewer's Responses to Questions

**Comments to the Author**

1. Is the manuscript technically sound, and do the data support the conclusions?

Reviewer #1: Partly

Reviewer #2: Yes

2. Has the statistical analysis been performed appropriately and rigorously? 

Reviewer #1: Yes

Reviewer #2: I Don't Know

3. Have the authors made all data underlying the findings in their manuscript fully available?

Reviewer #1: Yes

Reviewer #2: Yes

4. Is the manuscript presented in an intelligible fashion and written in standard English?

Reviewer #1: Yes

Reviewer #2: Yes

5. Review Comments to the Author

Reviewer #1: DiVincenzo and colleagues proposed a research article aimed at elucidating the role of miR-1469 in cutaneous melanoma unveiling its role in tumor invasiveness and in the development of ulcerating neoplasms. For this purpose, the authors have performed different functional experiments evaluating miR-1496 expression, cell function alterations and protein expression. Overall, the manuscript is interesting, however, it should be described better and some parts should be improved. Below are reported some comments:

1) Please provide references supporting the following sentences: “Cutaneous melanoma can sometimes present with ulceration of the tumor surface. Tumor ulceration is defined as the full thickness loss of epidermis overlying the tumor to the level of the basement membrane, with evidence of a host inflammatory response and associated thinning, effacement, or reactive hyperplasia of the adjacent epidermis. Tumor ulceration has been correlated with both decreased disease-free and overall survival in melanoma patients with otherwise similar staging criteria. Therefore, recognition of gross or histologic evidence of ulceration in cutaneous melanoma tissue has been incorporated into melanoma staging to better stratify melanoma patients. While ulceration is a feature that provides prognostic information to clinicians, few studies have explored the molecular features that explain its poor prognosis. Therefore, further studies are needed to characterize the molecular pathways that drive the ulcerated phenotype.”. For this purpose, please see:

– PMID: 29532857

- PMID: 27123116

– PMID: 25220403

– PMID: 29461778

- PMID: 28350549

2) In the Introduction the authors state: “Reduced expression of miR-1469 was identified in our assessment of miR expression patterns in ulcerated melanomas when compared to non-ulcerated tumors.”. In the results section they describe again the results of miR-1469 obtained for ulcerated and non-ulcerated melanomas. Probably the authors referred to nanostring results as previous results obtained and qPCR results in the present study, however, they have to better clarify this in the Introduction section;

3) Please provide references for this statement: “miR-1469 has also demonstrated consistent downregulated expression in other tumor types;”;

4) Please check the grammar in the following sentence: “however, a role for miR-1469 in melanoma has not been previously been reported.[7-10]”;

5) In the subheading “Patient samples and RNA isolation from formalin fixed paraffin embedded tissue”, How was the RNA extracted?;

6) Why the authors used hsa-miR-16 as endogenous control? For cell or tissue samples usually RNA U6 is generally preferred together with the analysis of an exogenous control like cel-miR-39. Please, clarify this aspect;

7) Were the data shown in Figure 1A obtained in previous studies? Please, clarify;

8) How do the authors explain the difference between the fold change value observed by Nanostring vs that obtained by qPCR (1.34 vs 11.81)?

9) Check the grammar of the following sentence: “As miR-1469 dysregulation was found to be a feature associated ulcerated primary cutaneous melanoma and limited studies have explored its role in cancer,”;

10) All these sentences should be moved into the Discussion section: “As a member of the BCL2 family of proteins, MCL1 is well recognized for its role as an inhibitor of apoptosis. Additionally, the expression of MCL1 has also been shown to affect the migratory and invasive capacity of tumor cells.[12] Increased MCL1 expression has also been demonstrated to occur in melanoma.[15] Furthermore, a recent study identified MCL1 as a confirmed target gene of miR-1469 in the context of laryngeal cancer by dual luciferase reporter assay.[9]”;

11) In the discussion section, the authors state :”However, the present results also stand in contrast to a recent study of miR-1469 as a promoter of cellular invasion in pancreatic cancer cell lines via inhibition of metastasis suppressor gene NDRG1 with subsequent activation of NF-κB.[25]”. Please better describe the role of NF-κB in melanoma invasiveness mediated by other factors including OPN and MMPs. For this purpose, please see:

– PMID: 28075446

– PMID: 32392801

Reviewer #2: The authors investigated the role of miR-1469 in ulcerated melanoma and found that this particular marker, if expressed, reduces the migratory and invasive capacity of melanoma cells in vitro, possibly by targeting MCL-1, a marker involved in apoptosis. The workflow is well thought and described and I believe the manuscript will have an impact in the field considering the molecular features associated with ulcerated melanoma that contribute to a poor prognosis are insufficiently explored.

However, minor revisions should be made and I advise the authors to spellcheck their manuscript.

The introduction section is too superficial. More information about miR-1469 and its roles should be given from the literature.

The number of patient samples included in this study should be given and information about patient informed consent should be included.

Statistical analysis is insufficiently described. Please describe the software used for statistical analysis and the algorithms used. Also, please indicate the number of experimental replicates used for the statistical analysis.

Line 45-54: a large section of the introduction is missing the reference(s).

Line 56, 66, 263, 307, 324, 360, 384: please check grammar

Line 70: the full name of the MCL-1 marker is not mentioned at all, only abbreviated, please amend this

Line 98: “24 hours or more”, please be more specific with the time frame.

Line 179, 372: it should be C instead of B, please amend this.

Fig 1C: How do the Authors explain the huge variation in data (standard deviation) in fold change of miR-1469 expression?

Fig.2A: The Authors are suggested to maintain the same scale for all 3 graphics, seeing as the values are similar.

Fig.2B: the scale bar is missing. The images corresponding to miR scramble in CHL1 and A375 cells seem to be insufficiently washed.

Fig.3B: the scale bar is missing.

Lines 249-250, 390-391: please check grammar, repetition of “following transfection”

Figure 4. Representative images should be provided for the trypan blue staining assay.

Line 269: “minimum p-value = 0.153, 0.117, and for CHL1”, I believe there is a missing value, please correct this.

Fig.5C: I suggest the authors place the cell lines’ names on the right of the corresponding immunoblot for an easier understanding and a better visual impact.

In the Discussion section I suggest the authors switch the paragraphs (326-332) and (334-348) with one another, as it will better link these portions with the rest of the text.

6. PLOS authors have the option to publish the peer review history of their article (what does this mean?). If published, this will include your full peer review and any attached files.

Reviewer #1: No

Reviewer #2: No

---

## [Author Response · Author response to Decision Letter 0]

9 Aug 2021

Reviewer #1.

Comment 1. “Please provide references supporting the following sentences: 

“Cutaneous melanoma can sometimes present with ulceration of the tumor surface. Tumor ulceration is defined as the full thickness loss of epidermis overlying the tumor to the level of the basement membrane, with evidence of a host inflammatory response and associated thinning, effacement, or reactive hyperplasia of the adjacent epidermis. Tumor ulceration has been correlated with both decreased disease-free and overall survival in melanoma patients with otherwise similar staging criteria. Therefore, recognition of gross or histologic evidence of ulceration in cutaneous melanoma tissue has been incorporated into melanoma staging to better stratify melanoma patients. While ulceration is a feature that provides prognostic information to clinicians, few studies have explored the molecular features that explain its poor prognosis. Therefore, further studies are needed to characterize the molecular pathways that drive the ulcerated phenotype.”. 

For this purpose, please see: PMID: 29532857, PMID: 27123116, PMID: 25220403, PMID: 29461778, PMID: 28350549.”

Response. We have added five additional references (PMID: 12932663, PMID: 7118080, PMID: 16595783, PMID: 29028110, PMID: 29850954) to these lines in order to support the Introduction section. 

Comment 2. “In the Introduction the authors state: “Reduced expression of miR-1469 was identified in our assessment of miR expression patterns in ulcerated melanomas when compared to non-ulcerated tumors.”. In the results section they describe again the results of miR-1469 obtained for ulcerated and non-ulcerated melanomas. Probably the authors referred to nanostring results as previous results obtained and qPCR results in the present study, however, they have to better clarify this in the Introduction section.”

Response. We agree with this comment regarding the difficulty in following the language used to refer to different experiments. Additional clarification has been provided in the introduction to amend this. “Reduced expression of miR-1469 was identified in our NanoString assessment of miR expression patterns in ulcerated melanomas when compared to non-ulcerated tumors.”

Comment 3. “Please provide references for this statement: “miR-1469 has also demonstrated consistent downregulated expression in other tumor types.”

Response. Citations have now been placed in the text to provide accurate references.

Comment 4. “Please check the grammar in the following sentence: “however, a role for miR-1469 in melanoma has not been previously been reported. [7-10]”

Response. We have changed the sentence to be, “However, neither a role for miR-1469 nor the effects of its reduced expression in melanoma have been previously reported.”

Comment 5. “In the subheading “Patient samples and RNA isolation from formalin fixed paraffin embedded tissue”, How was the RNA extracted?”

Response. We would like the clarify our methods for extracting RNA from formalin fixed paraffin embedded tissue. The Invitrogen RecoverAll Nucleic Acid Isolation kit was used and this has now been added to our materials and methods section. 

Comment 6. “Why the authors used hsa-miR-16 as endogenous control? For cell or tissue samples usually RNA U6 is generally preferred together with the analysis of an exogenous control like cel-miR-39. Please, clarify this aspect.”

Response. miR-16 was selected as an endogenous control based on recommendations by the manufacturer of the Taqman Advanced miRNA assay (ThermoFisher). miR-16 was identified as a recommended miR with stable expression across tissues that can function as an endogenous control in measurement of human tissue miR expression (PMID: 26921406, PMID: 17604727). A spike in exogenous control was not included in this assay as endogenous markers for normalization were available for application based on the source of RNA being tissue, rather than serum or plasma. 

Comment 7. “Were the data shown in Figure 1A obtained in previous studies? Please, clarify.”

Response. The Nanostring analysis data in Figure 1A was previously unpublished data from our group.[22] We understand the confusion regarding the language we used. In order to address this, we have removed the words “previously” and “first” from the results section entitiled “miR-1469 expression is significantly decreased in ulcerated cutaneous melanoma relative to non-ulcerated cutaneous melanoma”. 

Comment 8. “How do the authors explain the difference between the fold change value observed by Nanostring vs that obtained by qPCR (1.34 vs 11.81)?”

Response. One of the main differences between Nanostring and qPCR is the fact that Nanostring lacks an amplification step. Therefore, Nanostring analysis provides direct quantification of the number of each miR of interest in a sample and minimizes error that can come with additional sample handling in the conduct of molecular assays. The amplification step can incite bias within the sample in the form of the cDNA transcripts that are used for target detection by PCR. We feel this difference may explain the discrepancy. (PMID: 18278033, PMID: 28970855, PMID: 33627690)

Comment 9. “Check the grammar of the following sentence: “As miR-1469 dysregulation was found to be a feature associated ulcerated primary cutaneous melanoma and limited studies have explored its role in cancer.”

Response. We have added “with” to the sentence so that it now reads “As miR-1469 dysregulation was found to be a feature associated with ulcerated primary cutaneous melanoma…” to amend this. 

Comment 10. “All these sentences should be moved into the Discussion section: “As a member of the BCL2 family of proteins, MCL1 is well recognized for its role as an inhibitor of apoptosis. Additionally, the expression of MCL1 has also been shown to affect the migratory and invasive capacity of tumor cells.[12] Increased MCL1 expression has also been demonstrated to occur in melanoma.[15] Furthermore, a recent study identified MCL1 as a confirmed target gene of miR-1469 in the context of laryngeal cancer by dual luciferase reporter assay.[9]”

Response. We thank the reviewer for this suggestion. We agree that these sentences did not necessarily belong in the results section as they were written. In order to address this, we have modified this section and replaced the aforementioned sentences.

Comment 11. “In the discussion section, the authors state: “However, the present results also stand in contrast to a recent study of miR-1469 as a promoter of cellular invasion in pancreatic cancer cell lines via inhibition of metastasis suppressor gene NDRG1 with subsequent activation of NF-κB.[25]”. Please better describe the role of NF-κB in melanoma invasiveness mediated by other factors including OPN and MMPs. For this, please see: PMID: 28075446, PMID: 32392801.”

Response. In response to this comment, we have made changes to the Discussion section in order to further describe the role of NF-κB in melanoma invasiveness. To do this, the following was added: “While NF-kB activation has also been shown to promote metastasis and invasiveness in melanoma through downstream activation of several potentially tumorigenic factors including osteopontin (OPN) and matrix mellanoproteinases (MMPs) this opposing result of miR-1469 function in pancreatic cancer is most likely due to the diverse and cancer context-dependent effects of microRNA biology and thus these targets were not assessed in the current study.[31, 32]”

Reviewer #2.

Comment 1. “The introduction section is too superficial. More information about miR-1469 and its roles should be given from the literature.”

Response. We appreciate this comment and understand the need for additional information regarding miR-1469. To this point, we have amended our introduction to now include, “Reduced expression of miR-1469 was identified in our NanoString assessment of miR expression patterns in ulcerated melanomas when compared to non-ulcerated tumors. miR-1469 has also demonstrated consistently downregulated expression in other tumor types such as in lung, laryngeal and esophageal squamous cell cancers. [12-14] Furthermore, low expression of miR-1469 in esophageal squamous cell cancer was found to correlate with lymph node metastasis, tumor invasiveness and worsening disease progression.[14] Overexpression of miR-1469, on the other hand, was found to inhibit expression of myeloid cell leukemia-1 (MCL1) and consequently promote apoptosis in laryngeal cancer cells. Additionally, increased expression of the tumor suppressor gene p53 was found to promote miR-1469 production.[13] Thus, given that miR-1469 is not only involved in regulation of these few cancer-associated genes, but has a total of 75 possible target genes, it is likely that miR-1469 may be influential in tumorigenesis across many other cancer types as well.[15] While abnormal miR-1469 expression has indeed been identified in multiple malignancies, neither a role for miR-1469 nor the effects of its reduced expression in melanoma have been previously reported.”

Comment 2. “The number of patient samples included in this study should be given and information about patient informed consent should be included.”

Response. Patient sample numbers (ulcerated (n=13), non-ulcerated (n=11)) have been added to the “Patient samples and RNA isolation from formalin fixed paraffin embedded tissue” section of the Materials and Methods and the emphasis of patient consent under the approved IRB protocol (2007C0015) has also been supplemented in this section. 

Comment 3. “Statistical analysis is insufficiently described. Please describe the software used for statistical analysis and the algorithms used. Also, please indicate the number of experimental replicates used for the statistical analysis.”

Response. We appreciate this comment. We have amended the statistical analysis section in order to be more specific and detailed in our methods description. 

I added, “All statistics for in vitro data were performed using GraphPad Prism 9 statistical software. For all in vitro assays, statistical significance of differences between groups was analyzed using ANOVA or two-tailed Student’s t test and all data is based on 3 experimental replicates. Additionally, statistical software SAD 9.4 and R 3.6 was used for Nanostring data analysis with a fold change of at least 1.5 for any differentially expressed miRs identified. A p-value less than or equal to 0.05 was considered to be statistically significant. The Holm-Bonferroni method was used to adjust for multiple comparisons.” 

Comment 4. “Line 45-54: a large section of the introduction is missing the reference(s).”

Response. This comment has been addressed in response to reviewer #1.

Comment 5. “Line 56, 66, 263, 307, 324, 360, 384: please check grammar.”

Response. We have revised our grammar in these lines. 

Comment 6. “Line 70: the full name of the MCL-1 marker is not mentioned at all, only abbreviated, please amend this.”

Response. The full name of MCL-1 had been added to its first mentions in both the abstract (line 37) and introduction (line 76-77). 

Comment 7. “Line 98: “24 hours or more”, please be more specific with the time frame.”

Response. We understand the possible confusing nature of this language. We have removed the phrase “or more” from the manuscript in order to be as specific as possible with the 24-hour time frame.

Comment 8. “Line 179, 372: it should be C instead of B, please amend this.”

Response. We appreciate this being brought to our attention. “C” and been replaced with “B” in both instances. 

Comment 9. “Fig 1C: How do the Authors explain the huge variation in data (standard deviation) in fold change of miR-1469 expression?”

Response. The variation in data seen in Figure 1C was addressed in response to comment 8 from reviewer #1.

Comment 10. “Fig.2A: The Authors are suggested to maintain the same scale for all 3 graphics, seeing as the values are similar.”

Response. The Figure 2A scale bar on the y-axis of the A375 graph has now been increased to 200 to match the other 2 graphs. 

Comment 11. “Fig.2B: the scale bar is missing.”

Response. A 100-micron scale bar has now been added on to figure 2B. 

Comment 12. “The images corresponding to miR scramble in CHL1 and A375 cells seem to be insufficiently washed.”

Response. We understand it may appear due to the images that the miR scramble in CHL1 and A375 cells were insufficiently washed. However, a timed protocol was used for staining and washing of all inserts uniformly, and the images were analyzed using constant thresholds for elimination of background according to the method in our cited reference by Nyegaard et al. [23] Thus, we do not believe this affects our data interpretation.

Comment 13. “Fig.3B: the scale bar is missing.”

Response. A 100-micron scale bar has now been added on to figure 3B. 

Comment 14. “Lines 249-250, 390-391: please check grammar, repetition of “following transfection”

Response. This grammar issue has been resolved in both instances by changing the sentence to “MTS Proliferation assays performed at 24, 48, and 72 hours following transfection of melanoma cell lines with miR-1469 miR relative to scramble-transfected or untransfected cells.”

Comment 15. “Figure 4. Representative images should be provided for the trypan blue staining assay.”

Response. The representative images for trypan blue staining of the A375 cell line with three groups: transfection with negative control miR-scramble transfection with miR-1469-mimic, and no treatment have been included in supplemental figure 1 and are reflective of the values for viability obtained by trypan blue staining in figure 4A. 

Comment 16. “Line 269: “minimum p-value = 0.153, 0.117, and for CHL1”, I believe there is a missing value, please correct this.”

Response. Thank you for bringing this issue to our attention. We have addressed this by adding the additional minimum p-values: (minimum p-value = 0.153, 0.102, and 0.119 for CHL1, MEL39, and A375, Figure 5A and 5B).

Comment 17. “Fig.5C: I suggest the authors place the cell lines’ names on the right of the corresponding immunoblot for an easier understanding and a better visual impact.”

Response. We appreciate this suggestion for better figure interpretation and have made these adjustments to figure 5C. 

Comment 18. “In the Discussion section I suggest the authors switch the paragraphs (326-332) and (334-348) with one another, as it will better link these portions with the rest of the text.”

Response. We agree that this change improves the clarity of the Discussion as well and have thus switched these two paragraphs per your recommendation.

---

## [Editor Report · Decision Letter 1]

12 Aug 2021

Loss of miR-1469 expression mediates melanoma cell migration and invasion

PONE-D-21-15478R1

Dear Dr. Carson,

We’re pleased to inform you that your manuscript has been judged scientifically suitable for publication and will be formally accepted for publication once it meets all outstanding technical requirements.

Kind regards,

Klaus Roemer

Academic Editor

PLOS ONE
---

## [Editor Report · Acceptance letter]

24 Aug 2021

PONE-D-21-15478R1 

Loss of miR-1469 expression mediates melanoma cell migration and invasion 

Dear Dr. Carson III:

I'm pleased to inform you that your manuscript has been deemed suitable for publication in PLOS ONE. Congratulations! Your manuscript is now with our production department. 

Kind regards, 

on behalf of

Dr. Klaus Roemer 

Academic Editor

PLOS ONE